# Smokers' cognitive and behavioural reactions during the early phase of the COVID-19 pandemic: Findings from the 2020 ITC Four Country Smoking and Vaping Survey

**Shannon Gravely**[1]*, **Lorraine V. Craig**[1], **K. Michael Cummings**[2], **Janine Ouimet**[1], **Ruth Loewen**[1], **Nadia Martin**[1], **Janet Chung-Hall**[1], **Pete Driezen**[1], **Sara C. Hitchman**[3], **Ann McNeill**[3,4], **Andrew Hyland**[5], **Anne C. K. Quah**[1], **Richard J. O'Connor**[5], **Ron Borland**[6], **Mary E. Thompson**[7], **Christian Boudreau**[7], **Geoffrey T. Fong**[1,8]

1 Department of Psychology, University of Waterloo, Waterloo, Canada, 2 Department of Psychiatry & Behavioral Sciences, Medical University of South Carolina, Charleston, SC, United States of America, 3 Department of Addictions, Institute of Psychiatry, Psychology & Neuroscience, King's College London, London, England, 4 Shaping Public hEalth poliCies To Reduce ineqUalities and harM (SPECTRUM), Nottingham, United Kingdom, 5 Department of Health Behaviour, Roswell Park Comprehensive Cancer Center, Buffalo, NY, United States of America, 6 School of Psychological Sciences, Melbourne Centre for Behaviour Change, University of Melbourne, Melbourne, Australia, 7 Department of Statistics and Actuarial Science, University of Waterloo, Waterloo, Canada, 8 Ontario Institute for Cancer Research, Toronto, ON, Canada

* shannon.gravely@uwaterloo.ca

**Data Availability Statement:** All relevant data are within the paper. Requests for data used in this manuscript can be sent to: itc@uwaterloo.ca.

## Abstract

### Introduction

COVID-19 is primarily a respiratory illness, and smoking adversely impacts the respiratory and immune systems; this confluence may therefore incentivize smokers to quit. The present study, conducted in four high-income countries during the first global wave of COVID-19, examined the association between COVID-19 and: (1) thoughts about quitting smoking; (2) changes in smoking (quit attempt, reduced or increased smoking, or no change); and (3) factors related to a positive change (making a quit attempt or reducing smoking) based on an adapted framework of the Health Belief Model.

### Methods

This cross-sectional study included 6870 adult smokers participating in the Wave 3 (2020) ITC Four Country Smoking and Vaping Survey conducted in Australia, Canada, England, and United States (US). These four countries had varying responses to the pandemic by governments and public health, ranging from advising voluntary social distancing to implementing national and subnational staged lockdowns. Considering these varying responses, and the differences in the number of confirmed cases and deaths (greatest in England and the US and lowest in Australia), smoking behaviours related to COVID-19 may have differed between countries. Other factors that may be related to changes in smoking because of COVID-19 were also explored (e.g., sociodemographics, nicotine dependence, perceptions about personal and general risks of smoking on COVID-19). Regression analyses were conducted on weighted data.

**Funding:** Funding: This study was supported by grants from the US National Cancer Institute (P01 CA200512), the Canadian Institutes of Health Research (FDN-148477), the National Health and Medical Research Council of Australia (APP 1106451), and Health Canada's Substance Use and Addictions Program (SUAP) (2021-HQ-000058). GTF was supported by a Senior Investigator Award from the Ontario Institute for Cancer Research (IA-004) and the Canadian Cancer Society 2020 O. Harold Warwick Prize. RJO and AH are supported by a Tobacco Centers of Regulatory Science US National Cancer Institute grant (U54 CA238110).

**Competing interests:** Conflicts of Interest: KMC has served as paid expert witness in litigation filed against cigarette manufacturers. GTF has served as expert witnesses on behalf of governments in litigation involving the tobacco industry. AM is a UK National Institute for Health Research (NIHR) Senior Investigator. The views expressed in this article are those of the authors and not necessarily those of the NIHR, the UK Department of Health and Social Care, or Health Canada. KMC has served as paid expert witness in litigation filed against cigarette manufacturers. GTF has served as expert witness on behalf of governments in litigation involving the tobacco industry. AM is a UK National Institute for Health Research (NIHR) Senior Investigator. The views expressed in this article are those of the authors and not necessarily those of the NIHR, the UK Department of Health and Social Care, or Health Canada. All other authors have no conflicts of interest to declare. This does not alter our adherence to PLOS ONE policies on sharing data and materials.

## Results

Overall, 46.7% of smokers reported thinking about quitting because of COVID-19, which differed by country (p<0.001): England highest (50.9%) and Australia lowest (37.6%). Thinking about quitting smoking because of COVID-19 was more frequent among: females, ethnic minorities, those with financial stress, current vapers, less dependent smokers (non-daily and fewer cigarettes smoked/day), those with greater concern about personal susceptibility of infection, and those who believe COVID-19 is more severe for smokers. Smoking behaviour changes due to COVID-19 were: 1.1% attempted to quit, 14.2% reduced smoking, and 14.6% increased smoking (70.2% reported no change). Positive behaviour change (tried to quit/reduced smoking) was reported by 15.5% of smokers, which differed by country (p = 0.02), where Australia had significantly lower rates than the other three countries. A positive behavioural smoking change was more likely among smokers with: lower dependence, greater concern about personal susceptibility to infection, and believing that COVID-19 is more severe for smokers.

## Conclusions

Though nearly half of smokers reported thinking about quitting because of COVID-19, the vast majority did not change their smoking behaviour. Smokers were more likely to try and quit or reduce their smoking if they had greater concern about susceptibility and severity of COVID-19 related to smoking. Smokers in Australia were least likely to reduce or try to quit smoking, which could be related to the significantly lower impact of COVID-19 during the early phase of the pandemic relative to the other countries.

## Introduction

The coronavirus (COVID-19) pandemic is the greatest global threat from an infectious disease in over a century, and tobacco smoking is the world's leading cause of premature death from non-communicable diseases. Both smoking and COVID-19 are responsible for millions of deaths, with global smoking-attributable deaths mounting to more than 8 million people annually [1], and as of May 19, 2021, there have been about 163 million confirmed COVID-19 cases, and approximately 3.4 million people have died since it was declared a global pandemic on March 11, 2020 [2]. There is now some evidence that these two great threats are linked [3, 4]. While the World Health Organization (WHO) has identified those at increased risk of severe illness from COVID-19 to include people who are older (65+ years), have an underlying medical condition (e.g., lung and cardiac diseases), and/or a compromised immune system [5], tobacco smoking plays a major role in reducing lung capacity and increasing the risk and severity of many respiratory infections [3, 4, 6–8]. Moreover, smoking is the main common risk factor for many of the medical conditions that increase the risks of COVID-19 [5]. Because of this, many health authorities, including the United States (US) Centers for Disease Control (CDC) [8] and WHO [3, 4] have identified those who smoke (or have a history of smoking) as being at an increased risk for severe illness from COVID-19.

While the data are not conclusive, a number of observational studies have reported that smoking increases the risk of greater disease severity and mortality from COVID-19 [4, 9–13]. A recent meta-analysis, which included 40 studies, found that current smokers had a 58% increased risk of severe COVID-19 and a 46% increased risk of death from COVID-19 [13]. A

review of 32 peer-reviewed studies by the WHO concluded that although there were mixed findings (some studies reported no statistically significant association), the available evidence suggests that smoking is associated with increased severity of disease and death in hospitalized COVID-19 patients [4]. A cross-sectional population study of 53,000 adults in the United Kingdom (UK) reported that prevalence of COVID-19 was higher among current smokers than never-smokers [14]. In contrast, a population-level study conducted in 38 European countries (in May 2020) found that there was a significant negative association between smoking prevalence and the prevalence of COVID-19, and no direct association between smoking prevalence and COVID-19 mortality [15]. Regardless, smoking cessation is recommended by health authorities, clinicians, and public health organizations across the globe to reduce the risk of COVID-19 and to lessen its severity [3, 4, 8, 16–20]. The pandemic may be a unique time to do so [21].

The COVID-19 pandemic is a population-level stressor of unprecedented global proportions, and as a consequence has led to significant psychological trauma (e.g., stress, anxiety, depression, fear, and feelings of isolation and loneliness), as well as imposing sudden lifestyle changes and uncertainty about the future [22]. Therefore, understanding the interplay between COVID-19 and smoking behaviour is an important area of investigation. Because COVID-19 has created substantial psychological distress [22] and economic challenges [23, 24] throughout the world, it might be expected that the pandemic could result in increased smoking while trying to cope during these challenging times [25]. In contrast, given that smoking adversely impacts the respiratory [3, 6, 7, 19] and immune [6, 19, 26] systems, and COVID-19 is primarily a respiratory illness, some smokers may perceive COVID-19 as a threat to their health if they continue to smoke. Notably, an important question to investigate is whether smokers have been motivated to reduce or stop their smoking considering warnings on the potential perceived adverse effects of smoking on COVID-19 infection and illness outcomes, especially among smokers in countries with high case counts and death rates (e.g., the US and UK).

Several studies have examined the impact of COVID-19 pandemic on smoking behaviours across a range of countries (e.g., Australia, Canada, China, the Netherlands, Poland, Japan, Turkey, US, UK) [14, 27–45]. Many of these studies have reported varied behaviors among smokers (increasing or decreasing smoking, or no change in smoking), while others have focused on changes for a single outcome (e.g., quit attempts or cessation). While it is difficult to summarize and compare findings from the available literature due to differing methodologies between studies and inadequate reporting of the results in some cases, nonetheless, many of these studies have shown that a high proportion of smokers did not change their cigarette consumption during the pandemic [27–37]. Some studies have also reported that a similar proportion of smokers either increased or decreased smoking [27, 31, 32, 35], while others have found that there was a higher proportion of smokers who increased smoking than reduced smoking [13, 28, 29, 37]. Some studies only reported on either increasing [34, 38, 39] or decreasing [40, 41, 42] smoking, and these studies have shown highly varied estimates of either behavior.

Some studies have examined if smokers tried to quit and/or successfully quit, either during, or because of, COVID-19 [30, 34, 40, 43–45]. These studies have however reported highly variable estimates. A national population study in the UK of more than 10,000 respondents estimated that 2% (equivalent to 300,000 smokers) of UK smokers had quit smoking and 8% (equivalent to 550,000) had made a quit attempt as a result of COVID-19 [40]. Another national study in the UK reported that a similar proportion of smokers (12%) made a quit attempt because of COVID-19 [43]. A nationally representative study in the US reported that 26% of smokers tried to quit [34], and an online US study of dual users (who smoke and vape) found that 23% tried to quit smoking in order to the reduce risk of harm from COVID-19

[30]. A study from Turkey among smokers in a cessation clinic [44] and a UK national study [45] reported that there was an increase in quit attempts during the pandemic as compared to before it. In contrast, a report released in the US found that there has been a significant drop in smoking cessation Quitline calls during the pandemic compared to previous years [46].

An individual's perception of their risk of contracting COVID-19, and their likelihood of recovering from it, may motivate smokers to cut down or attempt to quit. Few studies have explored whether COVID-19 risk perceptions are related to changes in smoking. A national study in the UK showed that smokers who experienced significant stress about becoming seriously ill from COVID-19 were more likely to increase smoking [13], while other studies suggest that smokers who had greater concern of getting COVID-19 and/or those who perceive that smoking increases the risk of COVID-19, were more likely to be motivated to quit [31], reduce smoking [35, 42] or make a quit attempt [34]. While some of these studies appear to show that COVID-19 may incentivise some smokers to quit, they are somewhat limited by methodological shortcomings, such as the use of convenience samples, sampling bias (e.g., not representative of the smoking population), and/or not evaluating a range of other variables that may be related to changes in smoking. Therefore, more rigorous and robust studies are warranted. Moreover, utilizing a conceptual health behavior model can help guide such an evaluation.

The present study examined smokers' self-reported thoughts about quitting and changes in smoking during the early phase of the COVID-19 pandemic (April-June 2020) across representative samples of smokers from four high-income countries: Australia, Canada, England, and the US. To more clearly understand the relationship between COVID-19 and smoking behaviours, we adapted the Health Belief Model (HBM) [47] to conceptualize and explain individual changes in smoking behaviour. Notably, these four countries had varying responses to the pandemic, ranging from advising voluntary social distancing to implementing national and subnational staged lockdowns (Fig 1). Additionally, government messages about the pandemic were frequently misaligned with those from their own public health services in the US and UK, which appears to have fostered confusion about the severity of the pandemic as well as distrust among the public [48, 49]. Considering these varying governmental responses, and the differences in the number of confirmed cases and deaths (greatest in England and the US

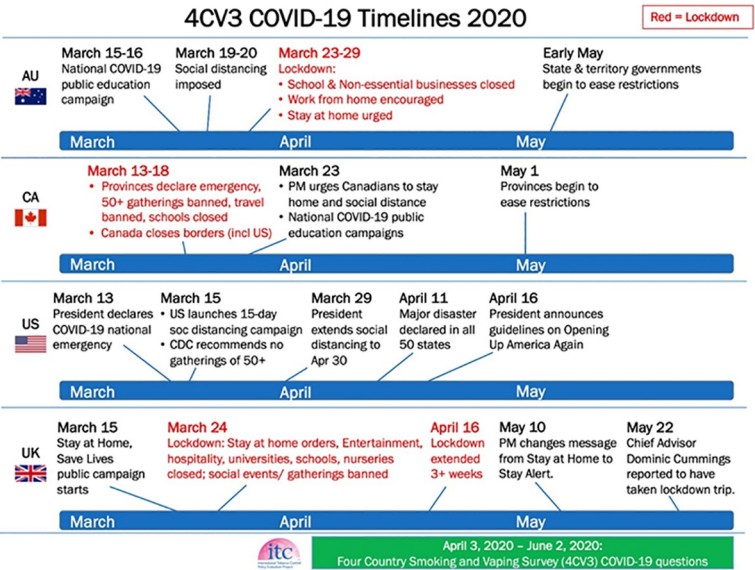

**Fig 1.**

and lowest in Australia), perceptions of risk and smoking behaviours related to COVID-19 may have differed between countries. The primary aims of this study were to examine the association between COVID-19 and: (1) thoughts about quitting smoking; (2) changes in smoking behaviour (quit attempts, reduced or increased smoking, or no change in smoking); and (3) factors related to a positive change (making a quit attempt or reducing smoking) based on an adapted framework of the HBM. We also examined: (4) what personal modifying factors (e.g., sociodemographics, mental health, nicotine dependence) were related to increased smoking because of COVID-19; (5) whether there were country differences among smokers and worry about getting COVID-19; and (6) whether those who tried to quit in the past month attributed their quit attempt to COVID-19. We hypothesized that because Australia was least affected, possibly owing to both being geographically isolated and the implementation of an early coordinated response by national, regional, and local governments that included a national lockdown, public adherence to the rules, and widespread testing [49], smokers in Australia would show lower levels of smoking-related cognitive and behavioural changes relative to smokers in the other three countries.

## Methods

### Study design, setting, participants

The ITC Four Country Smoking and Vaping (4CV) Survey is a longitudinal cohort study that consists of four parallel online surveys conducted in Australia, Canada, England, and the US. Adult ($\geq$18 years) respondents were recruited by online commercial panel firms in all four countries as cigarette smokers (smoked at least 100 cigarettes in their lifetime, and smoked at least monthly), recent ex-smokers (quit within $\leq$ 2 years), and at-least-weekly vapers or heated tobacco users. The sample in each country was designed to be representative of cigarette smokers, ex-smokers, and vapers (e.g., by age, sex, and region). Study questionnaires and materials were reviewed and cleared by Research Ethics Committees at the following institutions: University of Waterloo (Canada), King's College London (UK), Cancer Council Victoria (Australia), University of Queensland (Australia); and Medical University of South Carolina (waived due to minimal risk). All respondents provided written consent prior to completing the online survey, and all study procedures followed the principles of the Declaration of Helsinki. Further details about the sample and procedures can be found in the technical report [50].

Cross-sectional data for this study come from the 2020 4CV Survey (Wave 3). Survey fieldwork was conducted from February 24 to June 2, 2020, with a set of questions related to COVID-19 added to the survey on April 3, 2020. Thus, only the period of fieldwork from April 3 –June 2, 2020 included COVID-19 questions. Overall, 11607 respondents participated in Wave 3, of whom 9112 were current smokers ($\geq$ monthly). Eligible respondents included current smokers who: completed the survey from April to June (n = 7257, 79.6%) and had complete data for the outcomes assessed herein. We excluded the small number of recent ex-smokers who reported having quit smoking because of COVID-19 (n = 58), as they were not amenable to be included in the larger models. This resulted in a sample of 6870 smokers. A study flow diagram can be found in Fig 2.

### Measures

**Health Belief Model (HBM) framework.** The present analyses were conceptualized using an adapted HBM [47] which is commonly used for predicting individual changes in health behaviors. The model defines key factors that influence health behaviours and proposes that people are most likely to take preventative action if they perceive a specific health threat to be serious, and if they believe that they are personally susceptible to the threat [51].

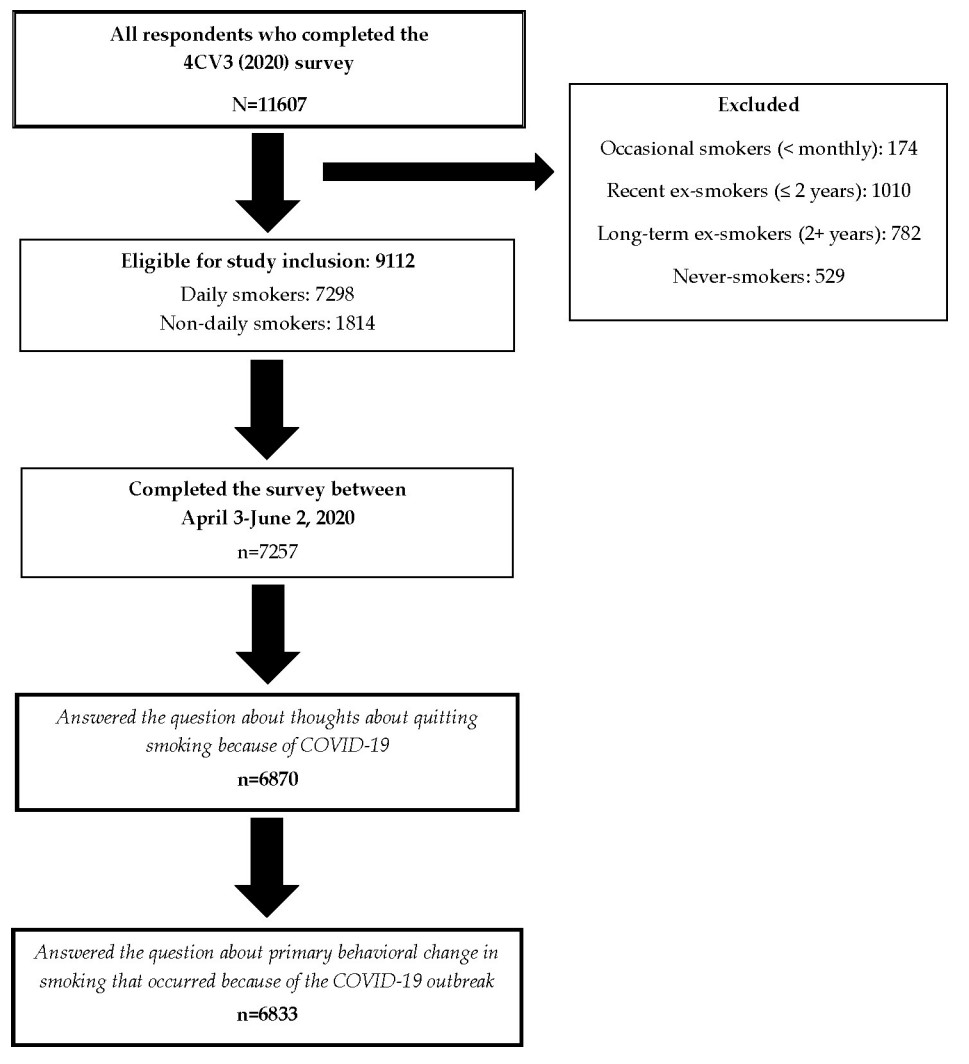

**Fig 2.**

In the present study, we included ten modifying factors (characteristics that may affect individuals' perceptions of their risk for COVID-19 and possible changes in smoking behaviour), and six personal threat variables related to smoking/COVID-19—four pertaining to 'susceptibility' (a subjective assessment of the risk of smoking and of developing COVID-19), and two pertaining to 'severity' (a subjective assessment of the severity of COVID-19, and its potential consequences). The three sets of variables are described below. The surveys and response options are available on the ITC Project website: https://itcproject.org/surveys/.

**Modifying factors.** Sociodemographic data were collected by the commercial panels and verified at the time of survey completion, including: age (18–39 vs. 40+), sex (male vs. female), ethnicity (ethnic majority: White/English vs. ethnic minority: Black/other minority), income (low, moderate, high, not reported), and education (low, moderate, high).

1. Financial stress: Respondents were asked, "In the last 30 days, because of a shortage of money, were you unable to pay any important bills on time, such as electricity, telephone or rent bills? Responses were coded as 'yes' vs. 'no'.

2. Smoking status: Smokers reported their smoking frequency at the time of the survey, and were categorized as daily or non-daily (weekly/ monthly) smokers.

3. Consumption: Smokers reported the number of cigarettes they smoked per day/week/ month, which was converted into cigarettes per day (CPD) for less than daily smokers and categorized as: 0–10, 11–20, 21+.

4. Vaping status: Respondents reported whether they were currently vaping at least monthly at the time of the survey. Responses were categorized into: 'current vaper' vs. 'non-vaper'.

5. Depressive symptoms: Respondents were classified as having depressive symptoms if they responded 'yes' to one of the following two questions: (1)"During the last 30 days, have you often been bothered by feeling down, depressed, or hopeless?" or (2) "During the last 30 days, have you often been bothered by little interest or pleasure in doing things?" Responses were coded as 'yes' vs. 'no/don't know'.

**Individual beliefs (threat variables).** *Personal susceptibility.*

1. All smokers were asked: To what extent, if at all, has smoking cigarettes damaged your health. Responses were categorized as: 'a great deal/a fair amount'; 'just a little' vs. 'not at all'.

2. All smokers were asked: How worried are you that smoking cigarettes will damage your health in the future? Responses were categorized as: 'very worried'; 'moderately/a little worried' vs. 'not at all worried'.

3. All smokers who had not had a positive COVID-19 test were asked: How worried are you that you already have, or that you will get the coronavirus? Responses were categorized as: 'extremely/very worried'; 'somewhat/a little worried' vs. 'not at all worried/don't know'.

4. All smokers were asked: Do you suspect you currently have the coronavirus, or that you had it and are now recovered? Responses were categorized as: 'Yes, I think that I have it now or have had it and recovered' vs. 'No, I do not think I have it now or had it in the past'.

*Personal severity.*

1. All smokers who had not had a positive COVID-19 test were asked: If you got the coronavirus, how severe do you think the illness would be for you, compared to others your age who got it? Responses were categorized as: 'a lot more severe'; 'somewhat/slightly more severe' vs. 'neither more or less severe/less severe/don't know'.

2. All smokers were asked: Thinking about smokers in general—if a smoker got the coronavirus, how severe do you think the illness would be for them, compared to non-smokers of the same age who got it? Responses were categorized the same as #1.

**Control variables.** The following were applied as control variables in the larger models: respondent type (cohort vs. replenishment), survey date (which were combined into the calendar week in which respondents completed their survey: example, 'week X: April 12–18, 2020'), and country of residence.

**Outcome measures.** *Impact of coronavirus on thinking about quitting.* All respondents were asked: "Did the coronavirus outbreak lead you to think about quitting?" Response options were: 'not at all', 'somewhat', 'very much', or 'don't know'. For the regression model,

these responses were dichotomized into: 'I thought about quitting smoking because of COVID-19' vs. 'I didn't think about quitting because of COVID-19'. The 'don't know' responses were included in the latter group.

*Smoking behaviour changes*. All respondents were asked: "What effect has the coronavirus outbreak had on your smoking?" Respondents could only select one response option. Responses were categorized first as: '(tried to) quit smoking', 'reduced smoking', increased smoking', 'no change in smoking', or 'don't know'. For the main regression models, this variable was dichotomized as: 'tried to quit/reduced smoking' vs. 'other response'. For the supplemental analysis, the outcome was 'increased smoking' vs. 'other response'.

**Statistical analyses.** Unweighted frequencies were used to describe the study sample. All other analyses were conducted on weighted data. In brief, a raking algorithm [52] was used to calibrate the weights on smoking status, geographic region, and demographic measures. Weighting also adjusts for the oversampling of vapers and younger respondents aged 18–24 recruited for this study.

## Cognitive and behavioural smoking responses to COVID-19

The first set of analyses included estimates of thoughts about quitting and behavioural responses related to COVID-19, overall and by country. An adjusted weighted regression analysis was conducted to compute estimates as to whether smokers thought about quitting because of COVID-19 (yes, very much; yes, somewhat; not at all; don't know, N = 6870). The between-country analyses compared thoughts about quitting (yes, very much/somewhat vs. otherwise). These analyses controlled for country and sociodemographic variables (age, sex, income, education, ethnicity).

A second adjusted weighted regression analysis estimated overall and by country, the primary smoking-related behavioural responses to the COVID-19 pandemic (tried to quit; reduced smoking; increased smoking; no change; don't know, n = 6833). The between-country analysis compared a positive behavioural change (tried to quit/reduced smoking) vs. otherwise (negative change/no change/don't know). These analyses controlled for country and sociodemographic variables (age, sex, income, education, ethnicity). Australia was used as the reference country due to the *a priori* hypothesis that smokers in Australia would be the least affected by the COVID-19 pandemic.

## Factors related to thinking about quitting smoking, making a quit attempt, or reducing smoking based on an adapted framework of the Health Belief Model

Two models were fit to examine smokers' behavioural responses using an adapted HBM framework. The first adjusted logistic regression model (Model 1) estimated the proportion of smokers who reported that they had thought about quitting smoking because of COVID-19 (yes, very much/somewhat vs. not at all/don't know). The second adjusted logistic regression model (Model 2) evaluated smokers' behavioural responses to COVID-19: positive change (tried to quit/reduced smoking) vs. otherwise (negative/no change/don't know). These two larger models included the modifying factors, as well as the susceptibility and severity variables. The models also controlled for country of residence, date of survey completion, and respondent type. Respondents who had already received a positive COVID-19 test at the time of data collection (n = 218) were excluded from the analytical models, as the questions 'worried they already had or would get COVID-19' and 'if you got the coronavirus, how severe do you think the illness would be for you, compared to others your age who got it' were not relevant.

### Modifying factors related to increased smoking because of COVID-19

An adjusted regression model was conducted to examine which modifying factors were related to increased smoking because of COVID-19. The outcome was increased smoking vs. other response.

### Worry about getting COVID-19

An adjusted multinomial regression model was conducted to examine if there were country differences by worry about getting (or already had) COVID-19. The outcome was: 'yes, extremely/very worried'; 'somewhat a little worried' vs. 'not worried at all/don't know'.

### Quit attempts because of COVID-19

The final analysis was conducted on weighted data (overall and by country) to estimate the proportion of smokers who attempted to quit within the last month prior to the survey (which is a conservative estimate of those who made a quit attempt during COVID-19), and subsequently estimated the conditional proportion of smokers who made that quit attempt because of COVID-19. Due to the small sample in the conditional quit category, the analysis only controlled for country, age, respondent type, and survey date.

All confidence intervals were computed at the 95% confidence level. Analyses were conducted using SAS Version 9.4 (SAS Institute Inc. 2013, Cary, North Carolina, USA).

## Results

Table 1 summarizes the characteristics of the smokers in the study sample.

### Thoughts about quitting smoking because of COVID-19

Table 2 summarizes thoughts about quitting smoking because of COVID-19 by country and overall. Among the 6870 smokers in this study, 46.7% reported that they have thought about quitting (very much/somewhat) because of COVID-19, which differed by country (p<0.001), with smokers in England reporting the highest rate (50.9%) and Australia the lowest (37.6%). Compared to Australia, smokers in Canada (47.9%, p<0.01) and England (p<0.001) were more likely to report having thought about quitting smoking because of COVID-19. The US did not significantly differ from Australia (43.3%, p = 0.10), but US smokers were less likely to have thought about quitting than those in Canada (p = 0.03) and England (p = 0.003).

### Behavioural responses to the COVID-19 pandemic

Among the 6833 smokers in this study who had complete data on their smoking-related behavioural reactions to COVID-19, 1.1% reported having tried to quit smoking, 14.2% reduced smoking, 14.6% increased smoking, 67.9% reported COVID-19 had no effect on their smoking, and 2.2% did not know. Positive changes in smoking (tried to quit or reduced smoking, 15.5% overall) differed by country (p = 0.02): smokers in the US (16.7%, p = 0.003), England (16.5%, p = 0.003), and Canada (16.0%, p = 0.004) were more likely to have tried to quit or reduce their smoking compared to smokers in Australia (9.3%) (Table 3).

### Analytical models adapted to the HBM framework

In the first analytical model (Model 1), modifying factors that were significantly associated with thoughts about quitting smoking were: female, ethnic minorities, non-daily smokers, current vapers, those who reported having financial stress, and lighter smokers (smoked fewer

**Table 1. Respondents' sample characteristics (unweighted %).**

| | Canada (n = 2613) | US (n = 1558) | England (n = 2097) | Australia (n = 602) | Overall (N = 6870) |
|---|---|---|---|---|---|
| **Respondent Type** | 38.5 | 40.4 | 3.9 | 5.7 | 25.5 |
| Cohort (recontact) | | | | | |
| New respondent | 61.5 | 59.6 | 96.1 | 94.4 | 74.5 |
| **Age** | 53.0 | 43.7 | 64.0 | 21.8 | 51.5 |
| 18–39 | | | | | |
| 40+ | 47.0 | 56.4 | 36.0 | 78.2 | 48.5 |
| **Sex** | 46.9 | 48.7 | 55.4 | 55.8 | 50.7 |
| Male | | | | | |
| Female | 53.1 | 51.3 | 44.6 | 44.2 | 49.3 |
| **Education** | 28.0 | 36.4 | 11.7 | 26.9 | 24.8 |
| Low | | | | | |
| Moderate | 44.2 | 41.1 | 54.2 | 41.4 | 46.3 |
| High | 27.8 | 22.5 | 34.1 | 31.7 | 28.9 |
| **Income** | 28.6 | 35.1 | 18.8 | 20.9 | 26.4 |
| Low | | | | | |
| Moderate | 27.9 | 27.9 | 26.2 | 20.4 | 26.7 |
| High | 37.6 | 37.0 | 48.9 | 52.3 | 42.2 |
| Not reported | 5.9 | 0.0 | 6.1 | 6.3 | 4.6 |
| **Financial stress** | 23.5 | 25.0 | 22.3 | 18.9 | 23.1 |
| Yes | | | | | |
| No | 75.5 | 73.4 | 76.5 | 80.9 | 75.8 |
| **Ethnicity** | 77.8 | 68.6 | 84.4 | 87.0 | 78.5 |
| Ethnic majority: White/English | | | | | |
| Ethnic minority: Black/other | 20.9 | 31.3 | 15.0 | 12.6 | 20.7 |
| Not reported/don't know | 1.3 | 0.1 | 0.7 | 0.3 | 0.8 |
| **Smoking Status** | 74.7 | 79.5 | 74.3 | 92.2 | 77.2 |
| Daily | | | | | |
| Non-daily | 25.3 | 20.5 | 25.8 | 7.8 | 22.8 |
| **CPD** | 59.6 | 56.8 | 63.3 | 40.4 | 58.4 |
| 0–10 | | | | | |
| 11–20 | 28.5 | 31.0 | 26.3 | 37.1 | 29.1 |
| 21+ | 8.8 | 8.6 | 5.0 | 20.1 | 8.6 |
| Don't know | 3.2 | 3.7 | 5.4 | 2.3 | 3.9 |
| **Vaping Status** | 37.4 | 29.8 | 48.5 | 16.0 | 37.2 |
| Current vaper | | | | | |
| Does not currently vape | 62.6 | 70.2 | 51.6 | 84.1 | 62.8 |
| **Depressive symptoms** | 56.5 | 40.8 | 53.2 | 47.4 | 51.1 |
| Yes | | | | | |
| No | 43.5 | 59.2 | 46.8 | 52.6 | 48.9 |
| **Positive COVID-19 test** | 2.3% | 5.3% | 2.5% | 3.8% | 3.2% |
| Yes | | | | | |
| No, negative | 1.5% | 0.6% | 0.5% | 0.3% | 0.9% |
| No result yet | 0.3% | 0.4% | 0.1% | 0.2% | 0.2% |
| Not tested | 95.9% | 93.6% | 96.9% | 95.7% | 95.6% |
| **Worried about getting COVID-19** | | | | | |
| Extremely/very | 21.7 | 21.6 | 32.7 | 17.7 | 24.6 |
| Somewhat/a little | 51.6 | 48.1 | 45.7 | 47.5 | 48.7 |

*(Continued)*

**Table 1.** (Continued)

| | Canada (n = 2613) | US (n = 1558) | England (n = 2097) | Australia (n = 602) | Overall (N = 6870) |
|---|---|---|---|---|---|
| Not at all | 25.5 | 28.8 | 19.7 | 33.0 | 25.2 |
| Don't know | 1.2 | 1.4 | 2.0 | 1.8 | 1.6 |
| **Suspected having had COVID-19** | | | | | |
| Yes | 10.4 | 14.3 | 21.7 | 3.7 | 14.1 |
| No/don't know | 89.6 | 85.7 | 78.3 | 96.3 | 85.9 |

CPD). With regard to the personal threat variables, the increased likelihood of having thoughts about quitting smoking was related to: smoking has caused damage to their health, worry that smoking will cause them health damage in the future, greater worry about getting COVID-19, if they suspected that they have had COVID-19, the belief that COVID-19 would be worse for them than others their age, and that COVID-19 is worse for smokers than for non-smokers. All weighted estimates, odds ratios, and 95% confidence intervals are presented in Table 4.

**Table 2. Thoughts about quitting smoking because of COVID-19 (weighted).**

| Response | Country | Mean | Lower CI | Upper CI |
|---|---|---|---|---|
| Very much | Australia[†] | 14.8% | 10.6% | 19.0% |
| | Canada | 20.9%* | 18.9% | 22.8% |
| | England | 28.6%* | 25.5% | 31.8% |
| | United States | 19.5% | 16.9% | 22.0% |
| | **Overall** | **21.9%** | **20.5%** | **23.3%** |
| Somewhat | Australia[†] | 22.6% | 17.7% | 27.5% |
| | Canada | 26.8%* | 24.6% | 29.0% |
| | England | 22.3% | 19.5% | 25.1% |
| | United States | 23.5% | 20.6% | 26.4% |
| | **Overall** | **24.5%** | **23.0%** | **25.9%** |
| Not at all | Australia[†] | 60.9% | 55.1% | 66.7% |
| | Canada | 50.8%* | 48.3% | 53.3% |
| | England | 46.3%* | 42.9% | 49.7% |
| | United States | 56.3% | 52.8% | 59.7% |
| | **Overall** | **52.0%** | **50.3%** | **53.7%** |
| Don't know | Australia[†] | 1.8% | 0.3% | 3.2% |
| | Canada | 1.6% | 1.0% | 2.1% |
| | England | 2.8% | 1.7% | 3.9% |
| | United States | 0.8%* | 0.4% | 1.3% |
| | **Overall** | **1.6%** | **1.2%** | **2.0%** |
| Yes, Very much/somewhat | Australia[†] | 37.6% | 32.0% | 43.5% |
| | Canada | 47.9%* | 45.4% | 50.4% |
| | England | 50.9%* | 47.5% | 54.3% |
| | United States | 43.3% | 40.0% | 46.8% |
| | **Overall** | **46.7%** | **46.7%** | **46.7%** |

N = 6870: Regression analysis outcomes: Very much: n = 1801; somewhat: n = 1758; not at all: n = 3157; don't know: n = 154; Combined categories: yes (very much/somewhat) vs. otherwise. Covariates included: age, sex, income, education, ethnicity.

Wald (logistic regression for combined categories): 8.1, p<0.0001. Country omnibus test: p = 0.0003.

[†]Reference

*significant at the p<0.05 level (compared to Australia).

**Table 3. Primary smoking behavioural reaction to COVID-19, overall and by country (weighted).**

|  | Canada | United States | England | Australia | Overall |
|---|---|---|---|---|---|
| **Tried to quit smoking** (n = 226) | 1.7% | 1.1% | 1.4% | 0.1% | 1.1% |
| p-value | < .0001 | < .001 | < .0001 | Reference |  |
| **Reduced smoking** (n = 1142) | 14.1% | 15.5% | 15.0% | 9.6% | 14.2% |
| p-value | 0.01 | 0.01 | 0.01 | Reference |  |
| **Increased smoking** (n = 1069) | 17.0% | 10.7% | 15.4% | 13.1% | 14.6% |
| p-value | 0.03 | 0.60 | 0.15 | Reference |  |
| **Don't Know** (n = 162) | 1.9% | 2.7% | 2.8% | 1.0% | 2.2% |
| p-value | 0.10 | 0.03 | 0.02 | Reference |  |
| **No effect on smoking**[*] (n = 4234) | 65.2% | 65.4% | 69.9% | 76.2% | 67.9% |
| **Tried to quit/reduced smoking**[†] (n = 1368) | 16.0% | 16.7% | 16.5% | 9.3% | 15.5% |
| p-value (vs. other response) | 0.004 | 0.003 | 0.003 | Reference | 0.022 |

N = 6833. Regression analysis outcomes

[*]Reference for outcome comparisons. Combined categories: tried to quit/reduced smoking vs. other response: increased smoking/no effect/don't know. Covariates included: country, age, sex, income, education, ethnicity.

[†]Country omnibus test: p<0.02.

In the second analytical model (Model 2), modifying factors that were related to greater likelihood of reporting a positive change (quit attempt or reduced smoking) included: smoking on a non-daily basis, and smoking fewer CPD. With regard to the personal threat variables, the following susceptibility and severity variables were associated with a greater likelihood of a positive behavioural change: those who perceive that smoking has damaged their health, greater worry about getting COVID-19, if they suspected that they have had COVID-19, and those who believe that COVID-19 is worse for smokers than for non-smokers. All weighted estimates, odds ratios, and 95% confidence intervals are presented in Table 5.

### Modifying factors related to increased smoking because of COVID-19

Those who were significantly more likely to increase smoking were: younger smokers (14.5% vs. older smokers: 12.2%), females (16.0% vs. males: 11.1%), those with financial stress (16.6% vs. those without financial stress: 12.3%), daily smokers (15.0% vs. non-daily smokers: 6.7%), and those with depressive symptoms (18.8% vs. those with no depressive symptoms: 9.0%) (Table 6).

### Smokers' worry about getting (or having had) COVID-19 by country

Smokers in England were more likely to report being extremely/very worried about getting or having had COVID-19 (31.5%), followed by smokers in Canada (18.6%), Australia (16.7%), and the US (15.2%). Australian smokers were more likely to report not being worried about getting COVID-19 (37.1%), followed by smokers in the US (32.1%), Canada (27.8%), and England (22.8%) (Table 7).

### Quit attempts due to COVID-19

Overall, 6.9% (weighted) of smokers who received the COVID-19 questions reported having tried to quit in the last month (n = 696/7257). The rate of making a quit attempt in the last month was the same as observed among smokers who completed the survey prior to the addition of the COVID-19 questions (n = 105/1855, 6.9% weighted, p = 0.53).

**Table 4. Model 1: Regression analyses examining thoughts about quitting smoking because of COVID-19.**

| Outcome 1: Because of COVID-19, I'm thinking of quitting smoking: yes, somewhat/very much | | Yes* weighted % | OR | 95% CI | |
|---|---|---|---|---|---|
| | | | | Lower CI | Upper CI |
| **Modifying Factors** | | | | | |
| Age | 18–39 | 46.9% | 1.2 | 1.0 | 1.4 |
| | 40+ | 43.4% | | Reference | |
| Sex | Male | 43.3% | **0.9** | **0.7** | **0.9** |
| | Female | 47.1% | | Reference | |
| Education | Low | 42.4% | 0.8 | 0.6 | 1.0 |
| | Moderate | 45.4% | 0.9 | 0.8 | 1.1 |
| | High | 47.9% | | Reference | |
| Income | Low | 43.9% | 0.9 | 0.8 | 1.1 |
| | Moderate | 45.4% | 1.0 | 0.8 | 1.2 |
| | High | 45.7% | | Reference | |
| Ethnicity | Ethnic majority | 43.9% | **0.8** | **0.6** | **1.0** |
| | Ethnic minority | 49.7% | | Reference | |
| Financial stress | Yes | 50.0% | **1.3** | **1.1** | **1.6** |
| | No | 43.7% | | Reference | |
| Smoking status | Daily | 43.8% | **0.7** | **0.6** | **0.9** |
| | Non-daily | 51.4% | | Reference | |
| CPD | 0–10 | 51.1% | **1.9** | **1.5** | **2.5** |
| | 11–20 | 37.3% | 1.1 | 0.8 | 1.4 |
| | 21+ | 35.3% | | Reference | |
| Vaping status | Current vaper | 53.6% | **1.5** | **1.3** | **1.7** |
| | Does not currently vape | 43.9% | | Reference | |
| Depressive symptoms | Yes | 43.9% | 0.9 | 0.8 | 1.1 |
| | No/don't know | 46.2% | | Reference | |
| **Personal Threat Variables** | | | | | |
| **Susceptibility** | | | | | |
| Has smoking cigarettes damaged your health? | | | | | |
| | A great deal/a fair amount | 48.3% | **1.4** | **1.1** | **1.7** |
| | Just a little | 44.6% | **1.2** | **1.0** | **1.4** |
| | Not at all//don't know | 40.9% | | Reference | |
| Worried smoking will damage your health in the future | | | | | |
| | Very worried | 52.1% | **2.1** | **1.6** | **3.0** |
| | Moderately/a little worried | 43.9% | **1.5** | **1.2** | **2.1** |
| | Not at all worried/don't know | 33.6% | | Reference | |
| Worried that you already have had, or that you will get the coronavirus | | | | | |
| | Extremely/very worried | 63.0% | **4.4** | **3.4** | **5.6** |
| | Somewhat/a little worried | 47.1% | **2.3** | **1.9** | **2.8** |
| | Not at all worried/don't know | 27.9% | | Reference | |
| Suspected having had COVID-19 | Yes | 53.6% | **1.5** | **1.1** | **1.9** |
| | No/don't know | 43.9% | | Reference | |
| **Severity** | | | | | |
| If you got the coronavirus, how severe do you think the illness would be for you, compared to others your age who got it? | | | | | |
| | A lot more severe | 55.7% | **1.2** | **1.0** | **1.5** |
| | Somewhat/a little more severe | 43.9% | 1.1 | 0.8 | 1.4 |
| | No difference/less severe/don't know | 37.9% | | Reference | |

*(Continued)*

**Table 4.** (Continued)

| Outcome 1: Because of COVID-19, I'm thinking of quitting smoking: yes, somewhat/very much | | Yes* weighted % | OR | 95% CI | |
|---|---|---|---|---|---|
| | | | | Lower CI | Upper CI |
| Severity of COVID-19 for smokers vs. non-smokers | | | | | |
| | A lot more severe | 55.7% | **2.1** | **1.6** | **2.7** |
| | Somewhat/a little more severe | 43.9% | **1.3** | **1.0** | **1.6** |
| | No difference/less severe/don't know | 37.9% | Reference | | |

N = 6635 had complete data for all variables; Data are weighted

*'Yes' is the main outcome presented in this table (yes: very much/somewhat = 3399 vs. otherwise (not at all/don't know) = 3236; CPD: Cigarettes smoked per day; CI: Confidence interval. Bolded data indicated statistical significant at the p<0.05 level.

Wald: 15.7, p<0.0001; Country: p = 0.20; Respondent type p = 0.47; Survey date: p = 0.001.

Among the smokers who completed the outcome question related to behavioural changes, 7.4% (weighted) made a quit attempt in the last month (n = 662/6833), with 6.0% (weighted) of those reporting that they did so because of COVID-19 (n = 66/662). This did not differ significantly by country (p = 0.19; England: 9.5%; Canada: 6.9%; US: 3.0%; Australia: 2.2%).

## Discussion

This study examined smokers' cognitive and behavioural reactions during the early phase of the COVID-19 pandemic, and found that nearly half the sample reported that they thought about quitting because of COVID-19. Smokers who had greater perceptions of general and personal susceptibility were more likely to have reported thoughts about quitting. However, very few smokers who thought about quitting because of COVID-19 actually made a positive behavioural change. These findings are in-line with Klemperer et al., where dual users in the US reported increased motivation to quit smoking if they perceived smoking increased their risk of harm from COVID-19, but most smokers did not change their smoking patterns [31]. Similarly, White et al. found that the majority of smokers believed that the risk of COVID-19 was greater for smokers than non-smokers [35]. And while higher perceived risk for COVID-19 for smokers was associated with a reduction in smoking, only a small proportion (23%) did so. These studies have shown that some smokers are concerned about COVID-19 and increased risk, and were motivated to stop smoking, but the pandemic does not seem to have carried enough weight to provoke actual positive changes.

While the current literature on changes in smoking behaviour has varied across studies (likely owing to methodological differences between studies), overall the international evidence appears to show that most smokers did not change their smoking patterns early on in the COVID-19 pandemic. Our findings support these results, with 70% of smokers reporting no change in their smoking, although our estimate of no-effect was higher than most others in the literature [28, 29, 31, 33], lower compared to a Canadian study [32], but equivalent to a study in the Netherlands conducted among Dutch smokers [27]. Our estimates of reduced smoking (14%) and quit attempts (1%) were much lower compared to some studies [31, 35, 40, 42, 43], but were very similar to others [27, 28–30, 41]. Studies varied widely in reporting of increased smoking, from 4% in Canada [32] to 50% in Australia [29], again likely because of differing samples and survey questions.

There appear to be some factors that are related to increased thoughts about quitting and changes in smoking. Our attempt to conceptualize various individual beliefs about smoking and COVID-19 showed that some smokers did report that they thought about smoking or reduced smoking or made a quit attempt because of COVID-19. Notably however, while we

**Table 5. Model 2: Regression analyses examining smokers' positive behavioural change in smoking (reduced or tried to quit smoking) that occurred because of COVID-19.**

| Outcome #2: Primary response: Tried to quit/reduced smoking | | Yes* weighted % | OR | 95% CI | |
|---|---|---|---|---|---|
| | | | | Lower CI | Upper CI |
| Modifying Factors | | | | | |
| Age | 18–39 | 13.4% | 1.2 | 0.9 | 1.4 |
| | 40+ | 11.7% | | Reference | |
| Sex | Male | 12.7% | 1.0 | 0.9 | 1.3 |
| | Female | 12.3% | | Reference | |
| Education | Low | 11.5% | 0.9 | 0.7 | 1.2 |
| | Moderate | 13.0% | 1.0 | 0.8 | 1.3 |
| | High | 12.7% | | Reference | |
| Income | Low | 12.3% | 1.0 | 0.8 | 1.2 |
| | Moderate | 12.2% | 1.0 | 0.8 | 1.2 |
| | High | 12.6% | | Reference | |
| Ethnicity | Ethnic majority | 12.1% | 0.8 | 0.7 | 1.1 |
| | Ethnic minority | 14.1% | | Reference | |
| Financial stress | Yes | 12.4% | 1.0 | 0.8 | 1.3 |
| | No | 12.5% | | Reference | |
| Smoking status | Daily | 11.0% | **0.4** | **0.3** | **0.5** |
| | Non-daily | 23.1% | | Reference | |
| CPD | 0–10 | 18.0% | **4.6** | **2.8** | **7.5** |
| | 11–20 | 8.9% | **2.1** | **1.2** | **3.4** |
| | 21+ | 4.5% | | Reference | |
| Vaping status | Current vaper | 14.3% | 1.2 | 1.0 | 1.5 |
| | Does not currently vape | 12.3% | | Reference | |
| Depressive symptoms | Yes | 13.0% | 1.1 | 0.9 | 1.3 |
| | No/don't know | 12.0% | | Reference | |
| **Personal Threat Variables** | | | | | |
| Susceptibility | | | | | |
| Has smoking cigarettes damaged your health | | | | | |
| | A great deal/a fair amount | 13.4% | **1.4** | **1.1** | **1.8** |
| | Just a little | 13.6% | **1.4** | **1.1** | **1.8** |
| | Not at all//don't know | 10.0% | | Reference | |
| Worried smoking will damage your health in the future | | | | | |
| | Very worried | 12.5% | 1.1 | 0.7 | 1.8 |
| | Moderately/a little worried | 12.9% | 1.1 | 0.8 | 1.6 |
| | Not at all worried//don't know | 11.4% | | Reference | |
| Worried that you already have had, or that you will get the coronavirus | | | | | |
| | Extremely/very worried | 16.4% | **2.3** | **1.7** | **3.1** |
| | Somewhat/a little worried | 14.1% | **1.9** | **1.5** | **2.5** |
| | Not at all worried/don't know | 7.9% | | Reference | |
| Suspected having had COVID-19 | Yes | 17.9% | **1.6** | **1.2** | **2.2** |
| | No//don't know | 11.9% | | Reference | |
| Severity | | | | | |
| If you got the coronavirus, how severe do you think the illness would be for you, compared to others your age who got it? | | | | | |
| | A lot more severe | 13.2% | 1.1 | 0.8 | 1.6 |
| | Somewhat/a little more severe | 13.3% | 1.1 | 0.9 | 1.4 |
| | No difference/less severe/don't know | 11.9% | | Reference | |

*(Continued)*

**Table 5.** (Continued)

| Outcome #2: Primary response: Tried to quit/reduced smoking | | Yes* weighted % | OR | 95% CI | |
|---|---|---|---|---|---|
| | | | | Lower CI | Upper CI |
| Severity of COVID-19 for smokers vs. non-smokers | | | | | |
| | A lot more severe | 14.7% | **1.6** | **1.2** | **2.3** |
| | Somewhat/a little more severe | 13.3% | **1.5** | **1.1** | **1.9** |
| | No difference/less severe/don't know | 9.5% | | Reference | |

N = 6617 had complete data for all variables. Data are weighted

* Made a quit attempt/reduced smoking is the main outcome presented in this table (made a quit attempt/reduced smoking = 1298 vs. otherwise: had thoughts about quitting/increased smoking/no change in smoking/don't know = 5319); CPD: Cigarettes smoked per day; CI: Confidence interval. Bolded data indicated statistical significant at the $p < 0.05$ level.

Wald: 10.1, $p < 0.0001$; Country: $p = 0.026$; Respondent type $p = 0.13$; Survey date: $p = 0.27$

found that nearly half the sample reported that they thought about quitting because of COVID-19, the majority did not try to stop. This was also reflected in the US study of dual users who reported increased motivation to quit smoking if they perceived smoking increased their risk of harm from COVID-19, but most smokers did not change their smoking patterns [31]. Both of these studies have shown that smokers were concerned about COVID-19 and increased risk, but it appears that the pandemic did not carry enough weight to provoke actual quit attempts. Additionally, while we found that only a small proportion of the sample reduced smoking or made a quit attempt because of COVID-19, individual's beliefs were strongly

**Table 6. Personal modifying factors related to increased smoking because of COVID-19.**

| Outcome 1: Because of COVID-19, I am smoking more (increased) | | Yes* weighted % | OR | 95% CI | |
|---|---|---|---|---|---|
| | | | | Lower CI | Upper CI |
| **Modifying Factors** | | | | | |
| Age | 18–39 | 14.5% | 1.22 | 1.01 | 1.49 |
| | 40+ | 12.2% | | Reference | |
| Sex | Male | 11.1% | 0.65 | 0.54 | 0.79 |
| | Female | 16.0% | | Reference | |
| Education | Low | 10.8% | 0.82 | 0.62 | 1.09 |
| | Moderate | 14.9% | 1.20 | 0.94 | 1.53 |
| | High | 12.8% | | Reference | |
| Income | Low | 14.3% | 1.09 | 0.85 | 1.40 |
| | Moderate | 12.4% | 0.93 | 0.73 | 1.17 |
| | High | 13.3% | | Reference | |
| Ethnicity | Ethnic majority | 14.0% | 1.30 | 0.99 | 1.70 |
| | Ethnic minority | 11.1% | | Reference | |
| Financial stress | Yes | 16.6% | 1.42 | 1.12 | 1.80 |
| | No | 12.3% | | Reference | |
| Smoking status | Daily | 15.0% | 2.45 | 1.85 | 3.23 |
| | Non-daily | 6.7% | | Reference | |
| Vaping status | Current vaper | 14.3% | 1.11 | 0.91 | 1.35 |
| | Does not currently vape | 13.1% | | Reference | |
| Depressive symptoms | Yes | 18.8% | 2.33 | 1.90 | 2.85 |
| | No/don't know | 9.0% | | Reference | |

Outcome: increased smoking vs. otherwise

**Table 7. Smokers' worry about getting (or having had) COVID-19 by country.**

|  |  | Canada | United States | England | Australia[†] |
|---|---|---|---|---|---|
| **How worried are you that you already have, or that you will get the coronavirus** |  |  |  |  |  |
| **Extremely/very worried** |  |  |  |  |  |
|  | Weighted % | 18.6% | 15.2% | 31.5% | 16.7% |
|  | OR, 95% CI | 1.5 (1.0–2.2)* | 1.1 (0.7–1.6) | 3.1 (2.0–4.7)*** | Reference |
| **Somewhat/a little worried** |  |  |  |  |  |
|  | Weighted % | 53.6% | 52.7% | 45.7% | 46.2% |
|  | OR, 95% CI | 1.5 (1.1–2.1)** | 1.3 (1.0–1.8) | 1.6 (1.2–2.2)** | Reference |
| **Not at all worried/don't know[†]** |  |  |  |  |  |
|  | Weighted % | 27.8% | 32.1% | 22.8% | 37.1% |

N = 6703. Regression analysis outcomes: Extremely/very worried: n = 1644; somewhat/a little worried: n = 3217; not at all worried/don't know:1788.

[†]Reference for outcome comparisons. Covariates included: country, age, sex, education, income, ethnicity, smoking status, CPD, respondent type and survey date.

*p<0.05

**p<0.01

***p<0.001 for country comparisons.

Wald (logistic regression for combined categories): 6.12, p<0.0001. Country omnibus test: p<0.001.

associated with positive changes in smoking behaviour. Specifically, smokers who reported greater perceptions about their personal (threat) susceptibility and severity for COVID-19 were more likely to have considered quitting and reduced or attempted to quit smoking, which is similar to studies in the US [29, 33, 43], but not consistent with the UK study that found worry about becoming seriously ill from COVID-19 was associated with increased smoking [13]. Therefore, for some smokers, the linkage between COVID-19 and smoking appears to have motivated cessation thoughts and behaviours. In contrast, for the many smokers who did not change their behaviour, or increased smoking, they may not have considered that smoking poses a COVID-19-related risk to them, and were not overly motivated to quit. In light of this finding, and despite the broad media coverage of cases and deaths across the globe, it is unclear whether smokers received strong public health messages about the potential effect of smoking on COVID-19 illness outcomes. It is also possible that due to the mixed evidence of whether it is riskier to smoke if COVID-19 was contracted [7, 14, 53, 54], coupled with suggestive evidence that that nicotine could in fact be a protective factor from COVID-19 [54–56], smokers may be confused about the role and potential impacts of nicotine and smoking on COVID-19. Some evidence has shown that messages on COVID-related risks and smoking may be effective in encouraging a quit attempt [57, 58], therefore stronger and accurate public health messaging should be disseminated.

With regard to modifying factors, we found few differences in positive behavioural changes. For example, we found that less dependent smokers were more likely to report a positive behavioural change, but there were no differences between any of the sociodemographic variables, vaping status, or depressive symptoms. In contrast, a UK study found that ethnic minorities and those who are more highly educated were more likely to have reduced smoking during the pandemic [13]. With regard to negative changes, a recent Canadian study found that people who reported financial difficulties during COVID-19, and have lower income and education, were more likely to increase tobacco use [32]. We did not find a relationship with education or income, but we did find that smokers with financial stress were more likely to increase smoking because of COVID-19. This finding is consistent with those of previous studies, in that, during financial difficulties, smokers will often shift their spending to tobacco and away from other critical necessities [59]. It is plausible that those who are experiencing stress and anxiety

due to the financial impacts of COVID-19 may be coping by increasing smoking. Our findings also suggest that highly dependent smokers (daily smokers), and other vulnerable groups, such as females and those with depressive symptoms also increased their smoking because of COVID-19. Heavier smokers and females were also found to be more likely to increase smoking in a UK study [13]. A recent national Canadian study also found that those with poorer mental health were also more likely to increase tobacco consumption during the early months of COVID-19 (March-April 2020) [32]. A recent qualitative study in the UK also found that smokers with mental health conditions and those with significant stress increased their smoking to help them cope during the pandemic [38]. The collective findings appear to show that there are some vulnerable groups who may be more susceptible to the effects of stressful life-events and cope by increasing smoking [13, 22, 29, 30, 32, 38].

In addition to individual differences, there were also some differences between countries. For example, smokers in Australia and the US were the least likely to think about quitting, while smokers in Canada and England had higher rates. Behavioural responses were less variable between countries, with the exception of Australia. There was no difference in the number of smokers who tried to quit or reduced smoking between the US, England, and Canada—with about 1 in 6 doing so—whereas only 1 in 11 Australian smokers reported either of these behaviours. In contrast, increased smoking was higher in Canada and England than in Australia and the US.

While the possible explanations of country differences are speculative on our part, our hypothesis that Australia would have lower smoking-related cognitive and behavioural changes was demonstrated by our findings, again, possibly owing to both being geographically isolated and the implementation of an early coordinated response by national, regional, and local governments that included a national lockdown, public adherence to the rules, and widespread testing [60]. During the early phases of the pandemic, Canada was less affected than the UK and the US. All provinces and territories declared states of emergency by mid-March 2020, pre-empting such a measure at the federal level. The hardest-hit provinces implemented lockdowns with strong containment measures [61]. In the US, the reason for lower rates of thoughts about quitting because of COVID-19 relative to Canada and England—despite the massive increase in cases and deaths in a very short period—is unclear, but may be related to conflicting messages about the severity of COVID-19, and/or political affiliation (as there were differences in reactions between the conservatives and the liberals) [62–64]. Additionally, US state and local governments have varied widely on restrictions, such as mask rules and limits on social gatherings [65]. The lack of consistency possibly resulted in much skepticism about the significance of the pandemic and the risks related to COVID-19. Finally, England was hit hardest in the early days of the pandemic and was overwhelmed by cases and deaths prior to, and during the study period, so smokers in England may have feared COVID-19 to a greater degree than those in the other countries. Indeed, our findings showed that smokers in England had the highest rate of worry about getting COVID-19 which may have motivated smokers to think about quitting and making a quit attempt. Quit smoking campaigns (e.g., #QuitforCovid) in the UK may have also increased cessation motivation [66], whereas these types of campaigns were absent in the other three countries. Thus, campaigns and educational materials to increase awareness of the harms of tobacco and the benefits of quitting, as well as how to cope during times of significant stress are critical, particularly for individuals who are susceptible to changing smoking behaviors during times of extreme stress.

## Limitations

While this study included a large sample of smokers from four countries, some caution is warranted when interpreting the results. First, this study was cross-sectional and cannot be used

to infer causality. However, the survey questions assessing thoughts about quitting and related changes in smoking were intentionally very specific, therefore we have no reason to believe that those who endorsed COVID-19 as a reason for having thoughts about quitting, reducing smoking, or making a quit attempt were not directly related to COVID-19. Second, questions about COVID-19 were not added to the survey until the beginning of April 2020, thus we were unable to capture some early and possibly important changes that may have occurred in March 2020 when the WHO declared the COVID-19 pandemic. Third, few recontact (cohort) respondents from Australia and England completed the COVID-19 questions because most recontacts from these two countries were surveyed prior to April 2020. However, there were no differences in quit attempts (in the last month) between respondents who did, or did not, get the COVID-19 questions, nor were there differences in thoughts about quitting smoking because of health concerns. Fourth, we did not ask about a specific quit date because quit was measured with broad time categories in our survey. This meant we had to limit our analysis of quit attempts to the last month, which may underestimate reactions to COVID-19. Fifth, we were unable to include data for quitters (who attributed quitting to COVID-19) due to the small sample size (n = 58). Many of these recent ex-smokers herein may have quit prior to the pandemic, thus the estimate for reduced smoking/making a quit attempt is a slight underestimate of respondents who did so. Sixth, smoking behaviours were self-reported. As self-reported quit intentions and reports of quit attempts may be biased by social desirability and/or poor recall, caution is warranted. Finally, respondents were only permitted to select one option for their behavioural reaction related to COVID-19, but they could potentially have reacted in multiple ways, therefore smoking-related behavioural reactions may be underestimated for some of the outcomes.

## Conclusion

This study examined smokers' thoughts about quitting and their primary smoking behavioural reactions to COVID-19 during the early months of the pandemic. We found that nearly half of smokers reported thinking about quitting because of COVID-19, but the vast majority did not change their consumption. This shows that there is a gap between intentions to quit and actual behaviors in quitting smoking due to COVID-19. While only a small proportion of the sample reported a positive change, individuals' beliefs about COVID-19 and smoking were associated with smokers having made a quit attempt or having reduced smoking if they have greater concern about personal susceptibility of infection, and/or believe that COVID-19 is more severe for smokers. Overall, our findings appear to suggest that the global health and economic shock caused by COVID-19 did not result in net changes in smokers' behaviour in all four countries in the short term.

## Acknowledgments

The authors would like to acknowledge and thank all those who contributed to the International Tobacco Control Four Country Smoking and Vaping Survey (ITC 4CV) Survey: all study investigators and collaborators, and the project staff at their respective institutions. We acknowledge comments from Cynthia Callard, Physicians for a Smoke-free Canada; Rob Cunningham, Canadian Cancer Society; and Francis Thompson, HealthBridge on drafts of this paper.

## Author Contributions

**Conceptualization:** Shannon Gravely, K. Michael Cummings, Pete Driezen, Geoffrey T. Fong.

**Data curation:** Christian Boudreau.

**Formal analysis:** Shannon Gravely.

**Funding acquisition:** K. Michael Cummings, Ron Borland, Geoffrey T. Fong.

**Investigation:** Shannon Gravely, K. Michael Cummings, Pete Driezen, Geoffrey T. Fong.

**Methodology:** Shannon Gravely, K. Michael Cummings, Janine Ouimet, Ruth Loewen, Sara C. Hitchman, Ann McNeill, Anne C. K. Quah, Ron Borland, Mary E. Thompson, Christian Boudreau, Geoffrey T. Fong.

**Project administration:** Janine Ouimet, Ruth Loewen, Anne C. K. Quah, Ron Borland.

**Resources:** Lorraine V. Craig.

**Supervision:** K. Michael Cummings, Geoffrey T. Fong.

**Validation:** Shannon Gravely, Pete Driezen.

**Visualization:** Shannon Gravely, Lorraine V. Craig, K. Michael Cummings, Geoffrey T. Fong.

**Writing – original draft:** Shannon Gravely.

**Writing – review & editing:** Lorraine V. Craig, K. Michael Cummings, Janine Ouimet, Ruth Loewen, Nadia Martin, Janet Chung-Hall, Pete Driezen, Sara C. Hitchman, Ann McNeill, Andrew Hyland, Anne C. K. Quah, Richard J. O'Connor, Ron Borland, Mary E. Thompson, Christian Boudreau, Geoffrey T. Fong.

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
