## [Decision Letter · Decision Letter 0]

24 Mar 2021

PONE-D-20-40626

Smokers’ cognitive and behavioural reactions during the early phase of the COVID-19 pandemic: Findings from the 2020 ITC Four Country Smoking and Vaping Survey

PLOS ONE

Dear Dr. Gravely,

Thank you for submitting your manuscript to PLOS ONE. After careful consideration, we feel that it has merit but does not fully meet PLOS ONE’s publication criteria as it currently stands. Therefore, we invite you to submit a revised version of the manuscript that addresses the points raised during the review process.

This paper received a split review, but I am willing to give you a chance to submit a revision.  Please be sure to address all the criticisms of both reviewers.  Per PLOS ONE policy, the "novelty" criticism is not relevant, but anything you can do to stress the importance of your contributions would be worthwhile.

We look forward to receiving your revised manuscript.

Kind regards,

Stanton A. Glantz

Academic Editor

PLOS ONE

Journal Requirements:

"Ethics approval: Study questionnaires and materials were reviewed and provided clearance by Research Ethics Committees at the following institutions: University of Waterloo (Canada, ORE#20803/30570, ORE#21609/30878), King’s College London, UK (RESCM-17/18-2240), Cancer Council Victoria, Australia (HREC1603), University of Queensland, Australia (2016000330/HREC1603); and Medical University of South Carolina (waived due to minimal risk). ".   

3. Please provide additional details regarding participant consent.

In the ethics statement in the Methods and online submission information, please ensure that you have specified (i) whether consent was informed and (ii) what type you obtained (for instance, written or verbal, and if verbal, how it was documented and witnessed). If your study included minors, state whether you obtained consent from parents or guardians. If the need for consent was waived by the ethics committee, please include this information.

'Conflicts of Interest: KMC has served as paid expert witness in litigation filed against cigarette manufacturers. GTF has served as expert witnesses on behalf of governments in litigation involving the tobacco industry. AM is a UK National Institute for Health Research (NIHR) Senior Investigator. The views expressed in this article are those of the authors and not necessarily those of the NIHR, the UK Department of Health and Social Care, or Health Canada. All other authors have no conflicts of interest to declare.  '

a. Please confirm that this does not alter your adherence to all PLOS ONE policies on sharing data and materials, by including the following statement: "This does not alter our adherence to  PLOS ONE policies on sharing data and materials.” (as detailed online in our guide for authors http://journals.plos.org/plosone/s/competing-interests).  If there are restrictions on sharing of data and/or materials, please state these.

Please note that we cannot proceed with consideration of your article until this information has been declared.

7. Please amend your list of authors on the manuscript to ensure that each author is linked to an affiliation. Authors’ affiliations should reflect the institution where the work was done (if authors moved subsequently, you can also list the new affiliation stating “current affiliation:….” as necessary).

8. Please include a separate caption for each figure in your manuscript.

9. Please include your tables as part of your main manuscript and remove the individual files. Please note that supplementary tables (should remain/ be uploaded) as separate "supporting information" files.

10. Please include captions for your Supporting Information files at the end of your manuscript, and update any in-text citations to match accordingly. Please see our Supporting Information guidelines for more information: http://journals.plos.org/plosone/s/supporting-information

Reviewers' comments:

Reviewer's Responses to Questions

**Comments to the Author**

1. Is the manuscript technically sound, and do the data support the conclusions?

Reviewer #1: Yes

Reviewer #2: Yes

2. Has the statistical analysis been performed appropriately and rigorously? 

Reviewer #1: Yes

Reviewer #2: No

3. Have the authors made all data underlying the findings in their manuscript fully available?

Reviewer #1: Yes

Reviewer #2: Yes

4. Is the manuscript presented in an intelligible fashion and written in standard English?

Reviewer #1: Yes

Reviewer #2: Yes

5. Review Comments to the Author

Reviewer #1: This is a well-written manuscript showing findings from a cross-sectional study conducted on a sample of tobacco smokers on the potential impact of the coronavirus pandemic on smoking habit during the early phase of the COVID-19 era. Data are original, the issue is interesting and the conclusions are acceptable. Authors should consider the following minor points to improve the presentation of findings.

1) Please save room in the Abstract to clarify the level of the Covid-19 lockdown during the fieldwork in various countries.

2) This study has some limitations, partially addressed in the Discussion section. Although the ITC study is a longitudinal study, at the end within this manuscript Authors considered a cross-sectional analysis, only. Thus, this analysis has the limitations inherent to the cross-sectional studies. Authors “excluded the small number of ex-smokers who quit smoking during COVID-19”. It is not clear to me if this has been due to the fact that the study did not allow Authors to identify all subjects who quit smoking due to the COVID-19 pandemic. In any case, these analyses do not allow Authors to answer to the following research question: did the Covid-19 pandemic reduce or increase smoking prevalence and consumption? In my opinion, this is a limitation of this study.

Reviewer #2: This study examines changes in and reasons for changes in smoking and vaping during the COVID-19 pandemic across four countries. There are several issues with the paper, including those related to novelty, measures, and analyses.

Introduction

• The introduction needs focus and a stronger literature review. There are many studies examining changes in smoking behavior during the pandemic, and a few studies have examined changes in vaping. While these are reviewed, the review is brief and the addition that the current study brings other than across countries isn’t developed well. For example, even just indicating that few studies have examined changes in vaping should be stated (although there are some more recent studies that should be included). The expanse of variables included in studies examining changes in tobacco use during the pandemic isn’t stated as should be. For example, some studies have focused on changes in smoking at all, with many findings within the same publication often showing some smokers increasing, some decreasing and some showing no changes at all. Other studies have shown changes in the number or frequency of use. These findings should be reviewed more in the current paper.

• The paper examines factors associated with changes in smoking and vaping during the pandemic. There are several studies that have included and examined reasons for changes in smoking and vaping, including risk perceptions. These studies are not discussed in the paper, nor are the reasons for this paper articulated.

• The first paragraph of the introduction, while interesting, seems unnecessary and better for the discussion if at all.

• Similarly, much of the discussion in the introduction concerning reasons for including the 4 countries is more of a commentary and better belongs in the discussion. For example, describing that England was overwhelmed or messages being misaligned don’t belong in the introduction. Instead, simply indicating that the 4 countries had different responses and differing messages might contribute to differing perceptions and tobacco use behaviors is all that is needed.

• The introduction offers a hypothesis earlier in the introduction before the research questions are offered, and a second hypothesis later in the introduction. Further, there are 3 supplemental analyses that are interesting and should be included as part of the main questions (especially iii, which is really related to the Health Belief Model).

• Would be good to update the number of COVID-19 cases and deaths given that the numbers change so quickly

Methods

• The survey items are not well developed and worded. For example, “How worried are you that you already have, or that you will get the coronavirus” is “double-barreled” and might result in unreliable responses. People who already have COVID-19 might not be worried especially if they are not that sick; whereas those who don’t have it but are concerned/at-risk might be more worried. The outcome variable concerning thinking about quitting during the pandemic is also problematic, as the survey didn’t first ask about desire to quit before asking if a reason to want to quit has to do with the virus. It is possible that people were heading towards quit anyway. Also, including “don’t know” with “I didn’t think about it…” is an assumption that probably should not be made. Similar issues are seen with some of the other survey questions. Granted, the items cannot be changed, but these issues should be acknowledged.

Results

• In the first set of results, regarding thoughts about quitting, the narrative only discusses 3 of the 4 countries. The data for the US should be stated too.

• Unclear why the variables in the supplemental analyses are supplemental and not part of the main analyses? Understanding why people increased is also very important.

• The results don’t separate out changes in vaping versus smoking. Are there differences worth noting?

Discussion

• The main differences in this study compared to the many other studies examining changes in tobacco use during the pandemic have to do with the vaping findings and the 4 countries. However, very little of the discussion focuses on these more novel aspects of the data.

Figures

• Don’t think Figure 2 is necessary

6. PLOS authors have the option to publish the peer review history of their article (what does this mean?). If published, this will include your full peer review and any attached files.

Reviewer #1: **Yes: **Silvano Gallus

Reviewer #2: No

---

## [Author Response · Author response to Decision Letter 0]

13 Apr 2021

* all responses to the reviewers can be found in the attached document "Responses to Reviewers". All responses to the editor are in the attached cover letter.

---

## [Decision Letter · Decision Letter 1]

17 May 2021

Smokers’ cognitive and behavioural reactions during the early phase of the COVID-19 pandemic: Findings from the 2020 ITC Four Country Smoking and Vaping Survey

PONE-D-20-40626R1

Dear Dr. Gravely,

We’re pleased to inform you that your manuscript has been judged scientifically suitable for publication and will be formally accepted for publication once it meets all outstanding technical requirements.

Kind regards,

Stanton A. Glantz, PhD

Academic Editor

PLOS ONE

Additional Editor Comments (optional):

Reviewers' comments:

Reviewer's Responses to Questions

**Comments to the Author**

1. If the authors have adequately addressed your comments raised in a previous round of review and you feel that this manuscript is now acceptable for publication, you may indicate that here to bypass the “Comments to the Author” section, enter your conflict of interest statement in the “Confidential to Editor” section, and submit your "Accept" recommendation.

Reviewer #1: All comments have been addressed

Reviewer #2: All comments have been addressed

2. Is the manuscript technically sound, and do the data support the conclusions?

Reviewer #1: Yes

Reviewer #2: Yes

3. Has the statistical analysis been performed appropriately and rigorously? 

Reviewer #1: Yes

Reviewer #2: Yes

4. Have the authors made all data underlying the findings in their manuscript fully available?

Reviewer #1: Yes

Reviewer #2: Yes

5. Is the manuscript presented in an intelligible fashion and written in standard English?

Reviewer #1: Yes

Reviewer #2: Yes

6. Review Comments to the Author

Reviewer #1: (No Response)

Reviewer #2: The authors have done an outstanding job revising the manuscript. I have no further issues, and thank the authors for an important paper.

7. PLOS authors have the option to publish the peer review history of their article (what does this mean?). If published, this will include your full peer review and any attached files.

Reviewer #1: **Yes: **Silvano Gallus

Reviewer #2: No

---

## [Editor Report · Acceptance letter]

28 May 2021

PONE-D-20-40626R1 

Smokers’ cognitive and behavioural reactions during the early phase of the COVID-19 pandemic: Findings from the 2020 ITC Four Country Smoking and Vaping Survey 

Dear Dr. Gravely:

I'm pleased to inform you that your manuscript has been deemed suitable for publication in PLOS ONE. Congratulations! Your manuscript is now with our production department. 

Kind regards, 

on behalf of

Professor Stanton A. Glantz 

Academic Editor

PLOS ONE